# Reliability Evaluation of Board-Level Flip-Chip Package under Coupled Mechanical Compression and Thermal Cycling Test Conditions

**DOI:** 10.3390/ma16124291

**Published:** 2023-06-09

**Authors:** Meng-Kai Shih, Yu-Hao Liu, Calvin Lee, C. P. Hung

**Affiliations:** 1Department of Mechanical and Computer-Aided Engineering, National Formosa University, Huwei, Yulin 632, Taiwan; 2Advanced Semiconductor Engineering (ASE), Nanzih District, Kaohsiung 811, Taiwan

**Keywords:** FCBGA, solder joint, finite element method, mechanical loading, reliability

## Abstract

Flip Chip Ball Grid Array (FCBGA) packages, together with many other heterogeneous integration packages, are widely used in high I/O (Input/Output) density and high-performance computing applications. The thermal dissipation efficiency of such packages is often improved through the use of an external heat sink. However, the heat sink increases the solder joint inelastic strain energy density, and thus reduces the board-level thermal cycling test reliability. The present study constructs a three-dimensional (3D) numerical model to investigate the solder joint reliability of a lidless on-board FCBGA package with heat sink effects under thermal cycling testing, in accordance with JEDEC standard test condition G (a thermal range of −40 to 125 °C and a dwell/ramp time of 15/15 min). The validity of the numerical model is confirmed by comparing the predicted warpage of the FCBGA package with the experimental measurements obtained using a shadow moiré system. The effects of the heat sink and loading distance on the solder joint reliability performance are then examined. It is shown that the addition of the heat sink and a longer loading distance increase the solder ball creep strain energy density (CSED) and degrade the package reliability performance accordingly.

## 1. Introduction

Over the past few years, demand has risen for electronic devices with increasing Input/Output (I/O) density and enhanced computational capabilities to support high-performance computing (HPC) applications such as AI (Artificial Intelligence) and machine learning. Flip Chip Ball Grid Array (FCBGA) packaging, 2.5D/3D heterogeneous integration packages, and Fan-Out Chip on Substrate (FOCoS) are the mainstay of such devices [1,2,3,4]. FCBGA (see Figure 1a) is particularly common in telecom, workstation, and computer applications due to its superior bandwidth and electrical performance. However, as the speed and I/O count of high-power packages increase, thermal dissipation control and management poses a critical concern [5,6,7]. Most of the heat generated by the chip is transferred to the printed circuit board (PCB) [8]. FCBGAs are commonly attached to a metal cavity heat spreader through a thermal interface material (TIM) to extend the heat conduction area of the die and further improve the heat dissipation (see Figure 1b). However, the resulting high-performance FCBGA (HFCBGA) assembly suffers the problem of an increased junction-to-case thermal resistance (Rjc) [9], which increases the junction temperature on the die.

The heat dissipation performance of HFCBGA packages can be improved further yet through their integration with an external heat sink or an air fan (see Figure 2) to force the heat flow upward and convection cooling use to remove excess heat. However, the additional load imposed by the heat sink and fan increases the risk of solder joint cracking at the point where the package is mounted on the PCB (Figure 3 shows the occurrence of solder joint fracture in the HFCBGA package mounted on the PCB with a heatsink) [10]. To improve the packaging yield rate, it is therefore necessary to properly understand the board-level fatigue reliability of advanced high-power consumption packages under coupled mechanical load and thermal cycling conditions.

The problem of minimizing the junction temperature and enhancing the heat dissipation capacity of advanced packages has attracted significant attention in the literature [11,12,13]. Han et al. [14] conducted computational fluid dynamics (CFD) simulations to investigate the heat dissipation capability of a 30 × 25 mm^2^ 2.5D package with various heat sink types and found that the package with the optimal heat sink structure was capable of dissipating more than 100 W of power. Rajan et al. [15] studied the thermal management efficiency of high-power CPUs fitted with micro pin fins on the backside of the silicon and a microfluidic cooling system. The results showed that the thermal resistance of the chip was reduced by around 44% compared to that of a conventional cold-plate cooling system. Lin et al. [16] developed a two-phase immersion cooling system to enhance the high-power dissipation of a chip on wafer on substrate (CoWoS) HPC package and showed that the thermal design power reached as much as 900, W. Su et al. [17] utilized a three-dimensional (3D) finite-element model (FEM) to investigate the thermal resistance of a fan-out (FO) package with various graphite terminal interface materials (TIMs). The optimal heat dissipation performance was obtained using a TIM fabricated of directional graphite.

The form factors of advanced packaging are trending towards ever larger sizes in order to meet the requirements for higher IO densities and power consumption. Thermal cycle on board (TCoB) tests have emerged as an important tool for evaluating the solder joint reliability in order to improve the packaging yield rate and physical integrity of the final assembly. Several studies have employed FE models to examine the TCoB reliability of solder interconnects [18,19,20]. Shao et al. [21] used a three-dimensional (3D) digital image correlation (DIC) technique to measure the warpage of a 55 × 55 mm^2^ 2.5D lidded package, and then constructed a verified FE model based on the DIC measurements to analyze the board-level thermal reliability for various geometry factors and material properties. The results revealed that the solder joint reliability was determined mainly by the thickness of the lid, PCB, and substrate, with a thinner thickness being preferred in each case. Lau et al. [22] conducted a thorough experimental and numerical investigation to assess the TCoB solder joint performance of fan-out panel level packaging (FOPLP) with a redistribution layer (RDL)-first and die face-down configuration. It was found that cracking of the solder joint occurred mainly at the interface between the RDL layer and the bulk solder. Ahmed et al. [23] performed nonlinear FE analyses using a global and sub-model modeling technique to analyze and assess the effects of heat sink application and the PCB laminate material properties on the solder joint reliability of 2.5D lidless structures. The simulation results showed that the solder joint reliability increased without the heat sink, with a thinner PCB thickness, and lower coefficient of temperature expansion (CTE) of the PCB laminate. Ji and Chai [24] combined a 3D slice FE model and a robust design-of-experiment (DOE) approach to examine the TCoB solder joint reliability of a 45 × 51 mm^2^ high density fan-out wafer-level package (FOWLP). The authors additionally employed a design technology co-optimization method (DTCO) to identify the dominant factors affecting the fatigue life of the solder joints under TCoB testing. Yang et al. [25] used a validated 3D FE model to analyze the solder joint plastic work energy of a lidless 2.5D package with a heat sink in board-level thermal cycling tests. The results showed that the solder fatigue life increased as the pre-load force of the bolts and CTE mismatch between the substrate and PCB reduced.

As described above, a larger package size and lidless structure are commonly used to improve the heat dissipation performance of high-power consumption package assemblies. Moreover, heat sink devices may also be added to the assembly to further cool the package. However, with the larger package size and external load imposed by the heat sink, the TCoB solder joint reliability of the package is a critical issue. Consequently, the present study constructs a 3D numerical model of a lidless FCBGA package with a heat sink to examine the solder joint reliability of the assembly under thermal cycling tests (TCT). The validity of the FE model is confirmed by comparing the simulation results for the warpage of the FCBGA package substrate with the experimental measurements obtained using a shadow moiré system. Finally, the validated model is used to examine the effects of various external load conditions on the mechanical characteristics and reliability performance of the solder joints. The simulation results provide useful guidance for evaluating the solder joint reliability of FCBGA packages with heat sink assemblies under coupled mechanical compression and thermal cycling test conditions.

## 2. Board-Level Thermal Cycling Tests

### 2.1. Test Vehicle Structure

In the present study, we considered and fabricated a lidless FCBGA assembly. As shown in Table 1, the package had an overall size of 40 × 40 mm^2^ and a build-up layer substrate thickness of 1.2 mm. The die had dimensions of 15.5 × 15.5 × 0.78 mm^3^. Finally, the underfill had a thickness of 25 μm.

### 2.2. Thermal Cycling Tests

The test vehicle was mounted on a 300 × 160 × 2.35 mm^3^ PCB test board made of FR-4 with eight layers and was subjected to board-level TCT testing in accordance with the JEDEC (Joint Electron Device Engineering Council) and IPC (Institute of Printed Circuits) standards [26,27]. The test board footprint and layout are shown in Figure 4a. The die was attached to the test board using an array of Sn-Ag-Cu solder joints (Figure 4b), with a diameter of 0.6 mm and a standoff of 0.55 mm. The opening size on both the package side and the test board side was set to 0.475 mm. Additionally, the pitch between adjacent solder joints in the array was set as 1.0 mm. A compressive fixture (see Figure 5a) was designed to examine the compressive loading effect of the heat sink on the die assembly. As shown in Figure 5b, the fixture was used to apply a force of 40 N to the FCBGA package to mimic the loading effect of the heat sink. A gel type thermal interface material was applied to the top surface of the chip with a maximum thickness of 0.1 mm after the heat sink (i.e., compressive metal plate) was attached to the base plate via four corner screws. In performing the thermal cycling tests, the assembly was placed in a thermal cycling chamber capable of applying a temperature range from −70 °C to 180 °C with a maximum ramp rate of 15 °C per minute.

The JEDEC TCT test conditions are specified mainly in terms of the temperature range and the heating/cooling rate. In accordance with JESD22-A104E Test Condition G [27], the TCT in the present study was performed using a temperature range from −40 to 125 °C with a dwell/ramp time of 15/15 min and a cycling rate of one cycle every 1 h (see Figure 6). A daisy chain circuit design was implemented to interconnect each solder joint in the FCBGA assembly, enabling the measurement of electrical resistance in the sample. The resistance measurements were captured by an event detector data acquisition system and monitored in real time to assess the change in resistance of the test sample over the course of the TCT.

### 2.3. Thermal Cycling Test Results

Table 2 shows the TCT reliability results obtained for the packages with and without the heat sink, respectively. For all three samples, the number of cycles to first failure is significantly reduced when the heat sink is added to the assembly. In other words, the additional loading force applied by the heat sink has a significant adverse effect on the reliability of the package. Figure 7 presents cross-sectional optical images of the die and substrate edge solder joints on the test vehicle without a heat sink following TCT. As shown, the initiation and failure of cracks predominantly transpired in the upper region of the solder ball near the substrate side, primarily affecting the joints situated at the corners of the BGA. A similar solder joint failure location was also observed for the test vehicles with a heat sink. Figure 8 compares the fractured solder joints of the test assemblies without a heat sink (Figure 8a) and with a heat sink (Figure 8b), respectively. It is evident that the mechanical compression force imposed on the solder joints through the additional weight of the heat sink results in a more severe cracking of the solder balls in the corners of the array as the FCGBA assembly undergoes thermal cycling.

## 3. Coupled Mechanical Compression and Thermal Cycling Simulation

### 3.1. Finite-Element Modeling

Figure 9 illustrates the lidless FCBGA with heat sink assembly considered in the present simulations. As shown in Table 1, the FCBGA model consisted of an eight-layer build-up organic substrate with a thickness of 1.2 mm and a single die with a footprint of 40 × 40 mm^2^ and thickness of 780 μm. The package was mounted on the test board using 96 Sn–4 Ag–0.5 Cu solder alloy joints. The TIM between the die and the heat sink had a thickness of 0.1 mm, while the heat sink had a size of 300 × 160 × 5 mm^3^.

Simulations were performed to investigate the warpage of the FCBGA and the accumulated inelastic work in the critical solder ball joints during TCT, with various heat sink arrangements. Due to the bi-axial symmetry of the architecture, the simulations considered only one quarter of the 3D board-level model for computational simplicity (Figure 10a). As shown in Figure 10b, the sub-model of the global-local model consisted of a single pitch solder ball. To simplify the analysis, the solder balls in the global model were meshed using simple cubic elements, and the solder bump layers were considered to be homogeneous layers. In accordance with the experimental observations, the sub-model of the solder ball used to examine the fatigue behavior of the solder joints was assigned to the upper-right corner of the global model beneath the substrate, which is generally regarded as the most critical point in the entire solder ball layout. 

The FE mesh consisted of a total of 389,702 elements of various types (including hexahedral and tetrahedron elements ANSYS solid186 3D modeling of solid structures). The global-level model and sub-model had average FE mesh sizes of 0.2 mm and 0.06 mm, respectively. Due to the bi-axial symmetry of the assembly, the simulations applied no-displacement boundary conditions in the X- and Y-directions along the YZ and ZX symmetry planes, respectively (see Figure 11a). Furthermore, to prevent any rigid body motion, the center point of the lower surface of the PCB was constrained in the z-direction.

### 3.2. Material Properties

The thermomechanical properties of the different components of the model are summarized in Table 3. The table includes Young’s modulus (E), coefficient of thermal expansion (CTE), glass transition temperature (Tg), and Poisson’s ratio (ν). The Young’s modulus of the test board was calculated as *C_0_ + C_1_ * T*, where *T* is the temperature in Kelvin [28]. As shown in Figure 12, the elastic moduli of the underfill and substrate were temperature dependent and were determined via dynamic mechanical analysis (DMA). As mentioned above, the solder bumps and underfill were considered to be homogeneous materials with approximately equivalent thermomechanical properties based on the relative volume fraction of the solder bump and underfill materials [29,30].

Throughout the duration of the TCoB tests, the solder balls experienced a consistently high homologous temperature. Hence, the deformation kinetics of the balls were dominated by a steady-state creep behavior [31,32] described by the following generalized Garofalo-Arrhenius constitutive equation [33,34]:(1)$${\dot{\varepsilon }}_{cr}=C_{1}{\left[\sinh(C_{2}\sigma )\right]}^{C_{3}}\exp\left(-\frac{C_{4}}{T}\right)$$

In the equation, ${\dot{\varepsilon }}_{cr}$ and σ represent the steady-state creep strain rate and stress, respectively. Moreover, *C*_1_–*C*_4_ denote constants with their respective values provided in Table 4.

### 3.3. Verification of Simulation Model

The validity of the simulation model was evaluated by comparing the numerical results for the warpage of the lower surface of the substrate (see Figure 13), with the warpage measurements obtained using an Akrometrix shadow moiré non-contact optical measurement system with an accuracy of 1 μm in the vertical displacement direction. 

Figure 14a,b shows the experimental and simulation results for the FCBGA warpage contours, respectively. In the modeling, we consider that the temperature decreases from the stress-free temperature at 150 °C to room temperature. That is immediately after cure of the underfill. Both result sets indicate that the package exhibits concave warpage when at room temperature. Based on the experimental findings, the highest warpage is observed at the substrate center, reaching approximately 122 μm (4.8 mils) in magnitude. The simulation results predict a maximum warpage of 128 μm (5.04 mils). The two measurements deviate by less than 5%. Thus, the basic validity of the FE model is confirmed.

### 3.4. Solder Joint Reliability Analysis

The coupled mechanical compression and TCT reliability of the board-level FCBGA was simulated using ANSYS FE code under the thermal loading conditions specified by JEDEC standard G. To simulate the mechanical compression effect of the heat sink in the experimental tests (see Figure 11b), a loading force of 40 N was applied to the heat sink at a distance of 45 mm from the center point in the *x*-axis direction and 50 mm in the *y*-axis direction. During the simulations, it was assumed that the temperature would decrease from a stress-free temperature of 217 °C to room temperature after the solder joint reflow process.

It was shown in [4] that the creep strain energy density (CSED) provides a reasonable index for examining the risk of solder joint cracking. Figure 15a shows the simulation results for the CSED distribution of the FCBGA package at the end of the thermal cycling test. The results confirm that the maximum CSED in the FE model occurs in the solder joint near the right-hand corner of the package. The results of the sub-model simulation are depicted in Figure 15b. It is evident that the highest CSED (creep strain energy density) is concentrated in the upper region of the solder ball. The simulation results are thus consistent with the experimental observations in Figure 7 and Figure 8. Thus, the validity of the FE model is further confirmed. 

The stress/strain data obtained from the FE simulations are affected by the mesh density and element size effects in the FE model. Therefore, an element volumetric averaging technique was applied to reduce the solution sensitivity with regard to the mesh density [32], i.e.,
(2)$$\Delta W_{ave}=\frac{\sum \Delta W_{total}\cdot V}{\sum V}\;,$$
where $\Delta W_{total}\;$ is the total CSED accumulated per cycle in the region, and *V* is the volume of the region that covers an area of 1 mm from the upper surface of the solder ball. The fatigue resistance of the solder joint interconnections was then evaluated by calculating the average CSED increment between cycle-1 and cycle-2 in accordance with:(3)$$\Delta W={\left(W_{avg.}^{}\right)}_{2nd}-{\left(W_{avg.}^{}\right)}_{1st}\;$$

### 3.5. Effects of Heat Sink

Figure 16 shows the simulation results obtained for the average SED under three different heat sink conditions: (1) without a heat sink; (2) with a heat sink and a loading force at 45 mm from the center point in the x-direction; and (3) with a heat sink and a loading force at 90 mm from the center point in the x-direction. The corresponding average SED values are 0.23 MJ/m^3^, 1.75 MJ/m^3^, and 1.98 MJ/m^3^, respectively. In other words, the SED increases when adding a heat sink to the FCGBA assembly due to the additional mechanical loading force applied to the die. Moreover, the average SED also increases with an increased loading distance from the center of the heat sink due to the longer loading distance resulting in a greater bending moment.

## 4. Conclusions

This study has examined the thermal cycling solder interconnection reliability of a board-level lidless FCBGA assembly with a heat sink under JEDEC thermal cycle testing (TCT). A 3D FE computational model with an inelastic creep behavior of the solder joints has been used to evaluate the solder ball TCT reliability of the package with and without a heat sink, respectively. The validity of the FE model has been confirmed by comparing the simulation results for the out-of-plane deformation of the FCBGA package substrate with the experimental measurements. The validated model has been used to investigate the CSED of the solder joints under various heat sink and loading force conditions. The experimental and simulation results support the following conclusions:

The board-level TCT results have shown that fracture occurs at the outermost corner solder ball joint in the BGA irrespective of whether or not a heat sink is attached to the package. However, the severity of the solder ball cracking increases under the additional load imposed by the heat sink. Consequently, the number of cycles to first failure is significantly reduced.The CSED within the solder balls at the outermost corners of the BGA increases with both the addition of the heat sink to the FCBGA assembly and an increasing distance of the loading force position from the center of the heat sink. A greater loading distance increases the compressive force acting on the BGA as the result of the bending moment, and is thus detrimental to the TCT reliability of the FCBGA assembly.

Overall, the present results show that an external heat sink and a greater loading distance have a significant adverse effect on the solder joint reliability of FCBGA packages. However, the addition of a heat sink is often necessary to improve the heat dissipation performance of high-power advanced packages. Consequently, further studies on the impact of the heat sink on the reliability of high-power packages are required to investigate such issues as the geometry and assembly loading force.

## Figures and Tables

**Figure 1 materials-16-04291-f001:**
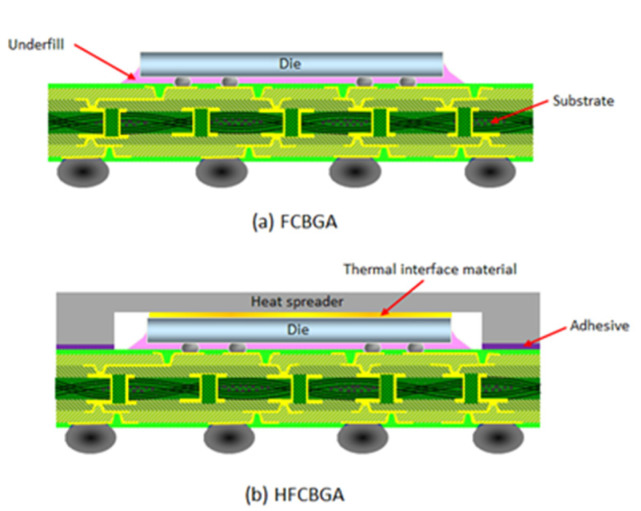
Schematic illustrations of: (**a**) FCBGA package and (**b**) HFCBGA package.

**Figure 2 materials-16-04291-f002:**
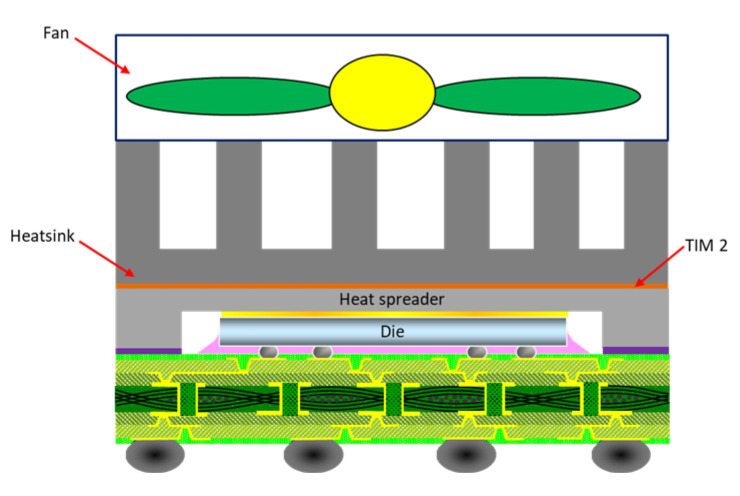
Schematic illustration showing HFCBGA with heat sink and air-fan assembly.

**Figure 3 materials-16-04291-f003:**
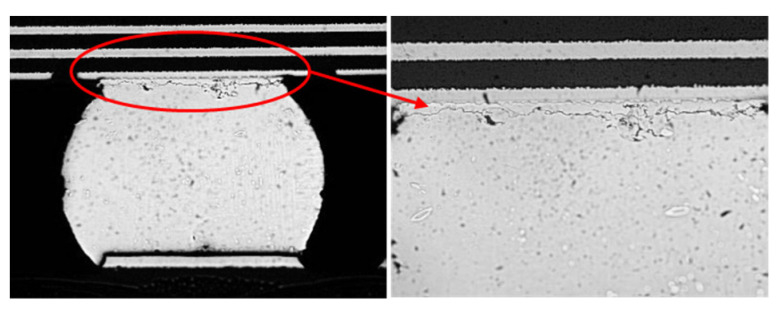
Cross-sectional images of solder joint crack.

**Figure 4 materials-16-04291-f004:**
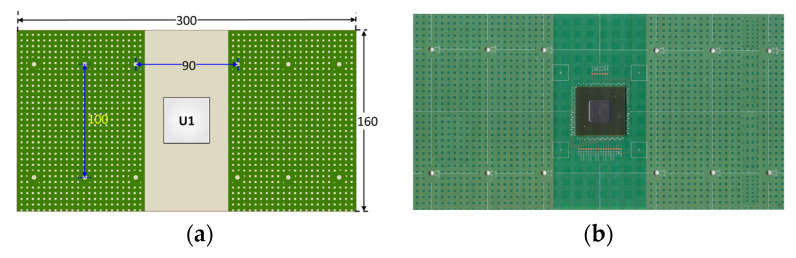
(**a**) Test board size and layout (Unit: mm) and (**b**) Lidless FCBGA package mounted on board.

**Figure 5 materials-16-04291-f005:**
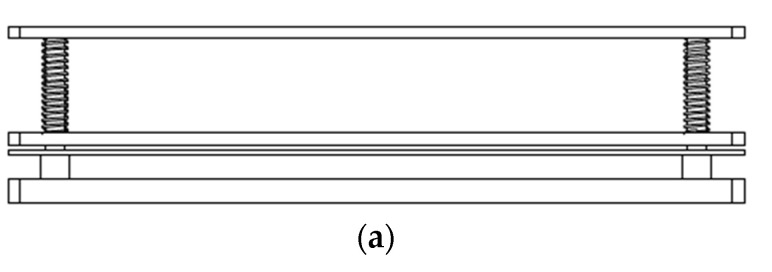
Schematic illustrations of (**a**) compressive fixture and (**b**) mechanical loading tests.

**Figure 6 materials-16-04291-f006:**
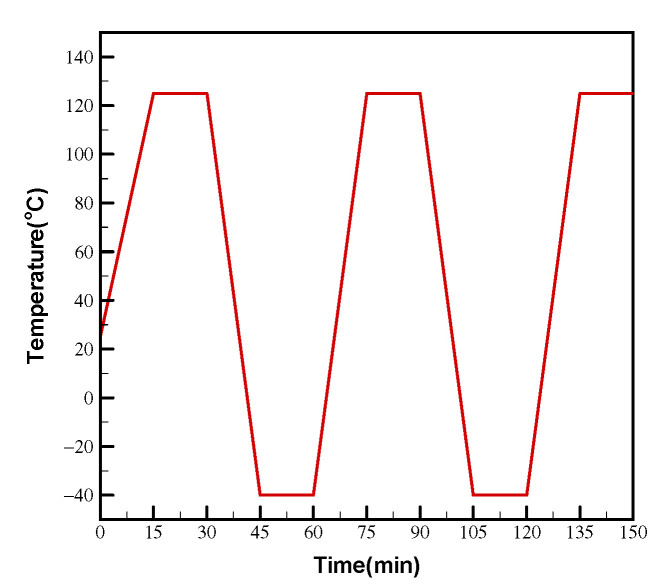
Thermal cycling temperature profile.

**Figure 7 materials-16-04291-f007:**
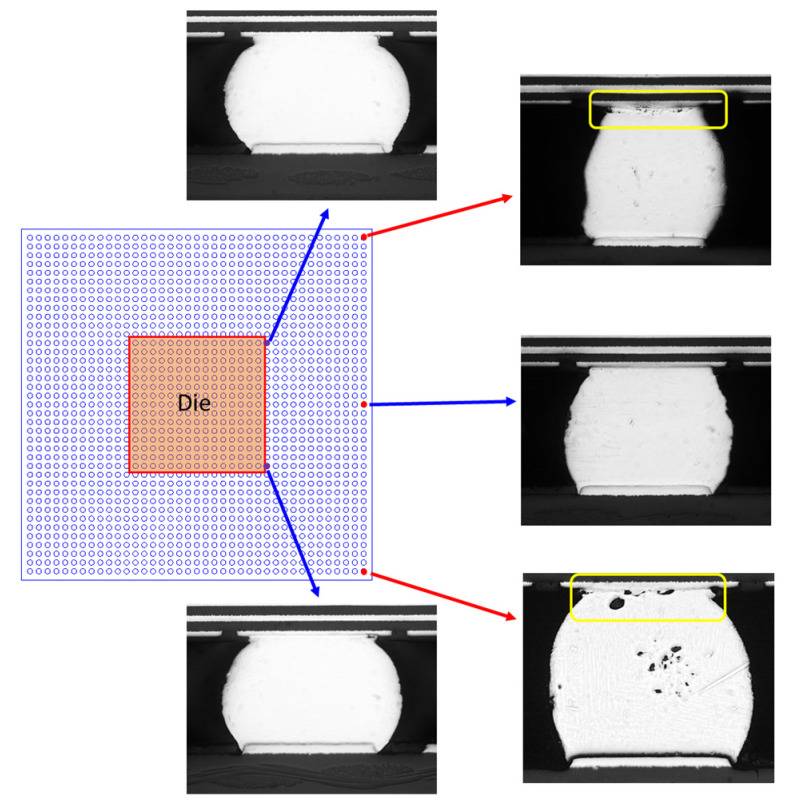
Cross-sectional views of die and package edge solder joints (w/o heat sink).

**Figure 8 materials-16-04291-f008:**
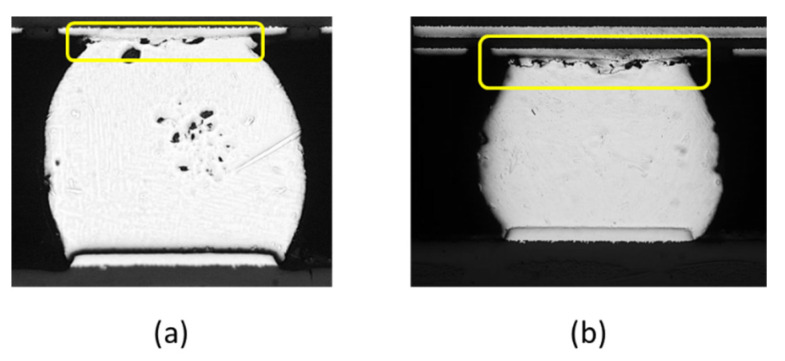
Cross-sectional views of fractured corner solder joints: (**a**) w/o and (**b**) w/heat sink.

**Figure 9 materials-16-04291-f009:**
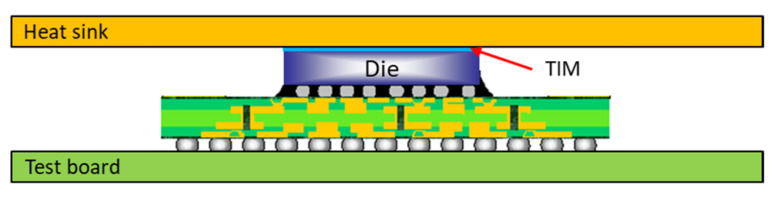
Schematic illustration of lidless FCBGA package with heat sink on test board.

**Figure 10 materials-16-04291-f010:**
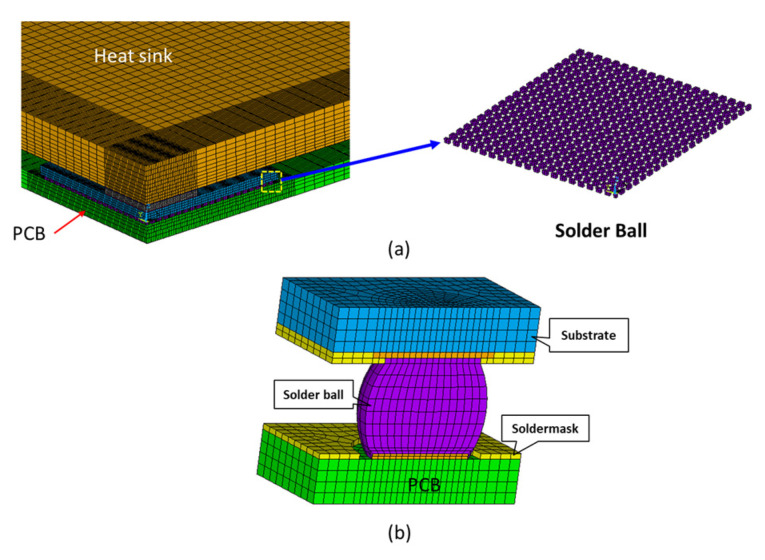
3D TCoB analysis model: (**a**) global-level model and (**b**) sub-model.

**Figure 11 materials-16-04291-f011:**
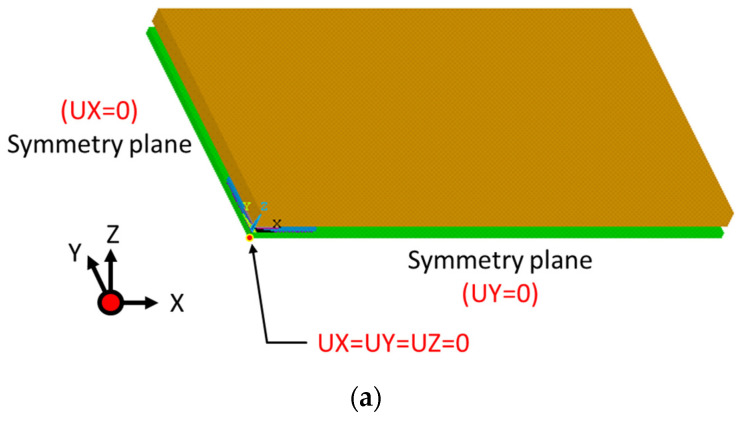
(**a**) Boundary conditions and (**b**) compressive loadng condition.

**Figure 12 materials-16-04291-f012:**
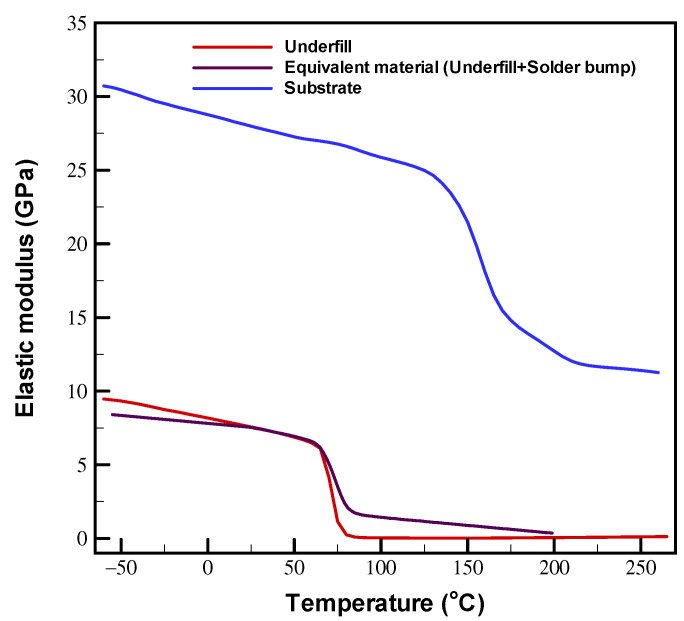
Temperature-dependent Young’s modulus of underfill and substrate.

**Figure 13 materials-16-04291-f013:**
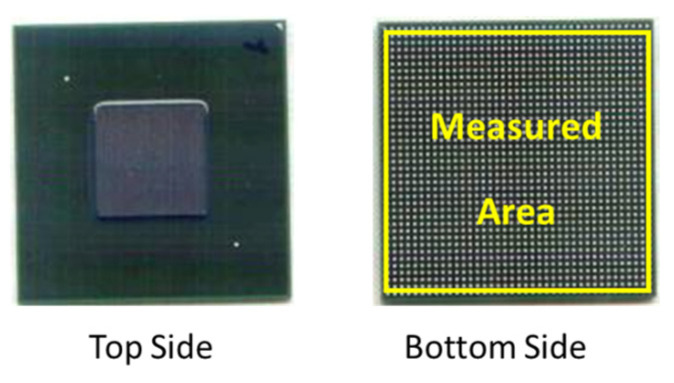
Warpage measurements of FCBGA package.

**Figure 14 materials-16-04291-f014:**
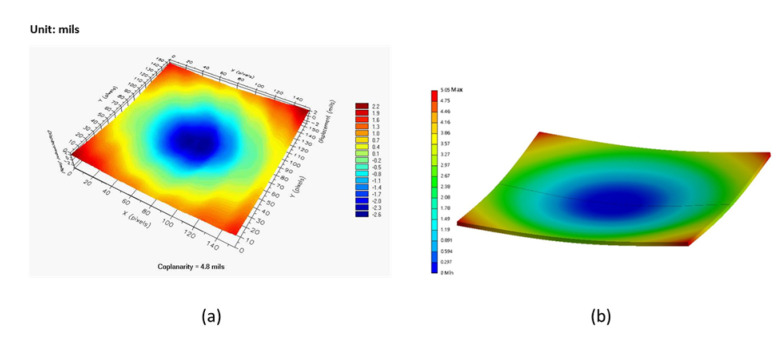
Warpage contour distributions of lower surface of substrate in lidless FCBGA package at 30 °C: (**a**) experimental results and (**b**) simulation results.

**Figure 15 materials-16-04291-f015:**
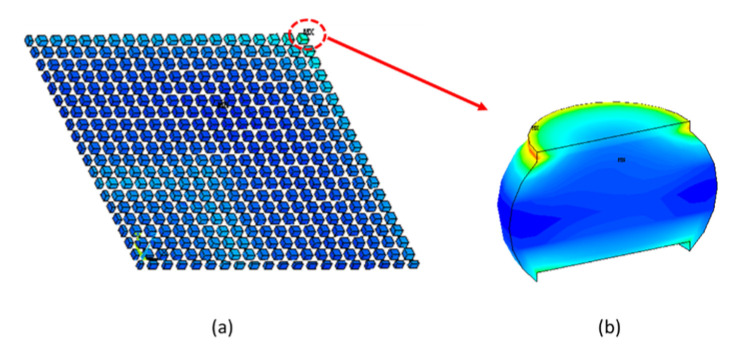
CSED distribution contours at 125 °C: (**a**) full package and (**b**) outermost solder ball.

**Figure 16 materials-16-04291-f016:**
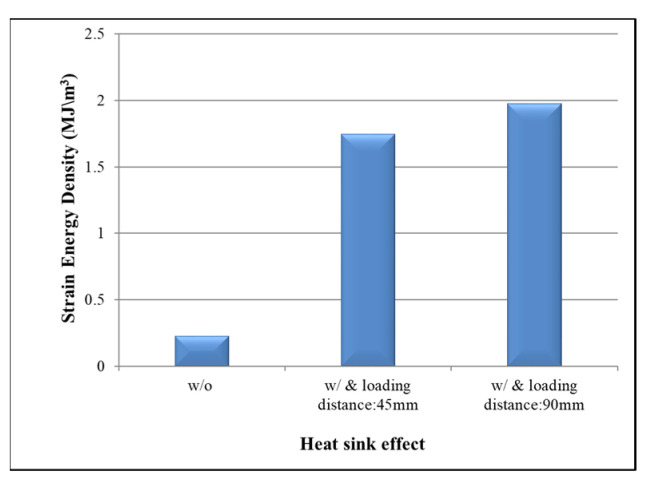
Average CSED in solder joints given different heat sink conditions.

**Table 1 materials-16-04291-t001:** Test Vehicle information.

Package size (mm)	40 × 40
Die size (mm)	15.5 × 15.5 × 0.78
Substrate thickness (mm)	1.2
Underfill thickness (μm)	25

**Table 2 materials-16-04291-t002:** TCoB test matrix and results.

Sample	Cycles to First Failure	
	W/O Heat Sink	W/Heat Sink
1	3652	2694
2	3689	2496
3	3865	2587

**Table 3 materials-16-04291-t003:** Thermomechanical properties of Package Components.

Component	E (GPa)	CTE (ppm/°C)	Tg (°C)	*ν*
Die	131	2.8	-	0.30
Solder mask	2.7	50/140	100	0.30
Underfill	Figure 12	32/110	70	0.30
Substrate	Figure 12	X/Y: 23.46 Z: 52.65/156.4	156	0.30
Test board	X/Y: *C*_0_ = 27.924, *C*_1_ = −0.0372 Z: *C*_0_ = 12.203, *C*_1_ = −0.016	X/Y: 16.5 Z: 67.2	-	0.39
Solder Ball	48.5@-55 °C 33@210 °C	20	-	0.35
TIM	0.0083	14	-	0.35
Heat spreader	68	24	-	0.344

**Table 4 materials-16-04291-t004:** Constants of Garofalo-Arrhenius Creep model for solder balls [34].

Constant	Unit	Value
*C*1	S^−1^	277,984
*C*2	MPa^−1^	0.02447
*C*3	n	6.41
*C*4	K	6489.7

## Data Availability

The data presented in this study are available on request from the corresponding author.

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
