# Peer review of "Reliability Evaluation of Board-Level Flip-Chip Package under Coupled Mechanical Compression and Thermal Cycling Test Conditions"

_materials, 2023, doi:10.3390/ma16124291_

Round 1
Reviewer 1 Report
The manuscript has sufficient interest. The paper makes an acceptable case for publication in “Materials”. To clarify both advantages and limitations of the study, the reviewer feels that justifications on the used techniques are enough. The authors made a fair-minded comparison with previous works and put emphasize on a clear scientific discussion in the paper.
As a result, the reviewer’s idea is that this manuscript should be accepted.
Author Response
Appreciate review’s recommendations. Your comments and corrections have made this manuscript more scientific.
Reviewer 2 Report
Why did you apply a vertical force of 40 N which corresponds to a weight of 4 kg? The weight of an actual heat sink would be much less and hence induce significantly lower additional stresses. The failure phenomenons you depict clearly indicates shear stresses to be main damage causes. It is not clear by how much the additional compressive stresses contribute to the damage evolution. FEA provides the possibility to evaluate not only the equivalent but the principal stresses too. Hence it might be interesting to invetigate those to explain in more detail the effect of the presence of a heat sink. The validation procedure is not described at an appropriate detail level. What temperature profile was applied in the measurement? What load profile was used for the simulation? Not only the deformation state at room temperature should be matching but also at higher (and lower) temperatures. The close match at room temperature has to be questioned since you did not consider viscoelastic behaviour of the PCB material. A validation at the test setup level is missing. Simulation considers the heat sink to be present during the solder reflow. Wouldn't the heat sink assembly take place after that? The Garofalo creep model you applied will result in extremly high creep strains at the high temperatures you considered in the simulation. The model was not meant for that use. You may add literature covering material characterisation and model determination work including temperatures as high as 220°C. Conisdering all elements that are within a distance of 1mm from the upper pad means that more or less the entire solder joint volume is considered. That way, no focus is done on the actually stressed part of the solder joint. Cycles 1 and 2 from FE simulations of TC tests usually show quite high differences in the creep energy or creep strain per cycle. It is common to at least calculate three cycles to reach a steady state behaviour also by means of the calculation method. Which cycle did you eventually use to compare the average creep energy per cycle with or without a heat sink? Yes, a greater distance causes an increased bending moment. However, this does not necessarilly correspond with a higher compressive stress in all solder joints. More detailled evaluation is needed here. A critical comparison of the observed cycles to failure and change in creep energy per cycle is missing. In general, the paper provides limited new knowledge and meets the requirements of a conference contribution. However, I suggest to not to accept it as a journal article.
Author Response
We thank you for your positive comments on our manuscript. Your comments and corrections have made this manuscript more scientific. We have revised our manuscript according to your comments and corrections.

Reviewer 3 Report
The paper uses FEA to investigate the solder joint reliability of a lidless on-board FCBGA package having heat sink effects under thermal cycling. The paper is quiet interesting and have the merits for publication. The following minor reviews are suggested:
1- Thermal reliability of solder interconnections is function of the inelastic strain energy density and plastic strains. Not on stresses. Because the deformations are highly dominated by creep. Please correct this in the abstract and elsewhere through the manuscript.
2-More details on the FEA model. Like element type used, mesh quality/density.
3- In fact, simulating two thermal cycles in FEA Is not recommended. At least, 3cycles are required. Better be, 5 complete thermal cycles.
4- Please compare results with literature, if possible.
5- To my best knwoledge, G-condition of JEDEC is 15 min dwell and 20 min ramp. The paper mentions 15/15 min dwell/ramp. Please check.
The paper is written with a high-quality english language.
Author Response

(The authors gave the same response as above.)

Reviewer 4 Report
This was well organized and showed a systematic study.
They address the question of whether an applied heat sink will increase the reliability. Normally, it is believed that the cooler the better but the imposed stress may be not beneficial for the internal solder joints in the Hybrid Integration.
The topic is fairly rare since this analysis seems pretty obvious but in fact the conclusion is not so obvious
The paper is well written, it is very clear and well organized
The conclusions consistent with the evidence and arguments presented. They answer with a nice experiment and the simulation fits very well to the data. It is a very good and pretty original with an important conclusion.
Author Response

(The authors gave the same response as above.)

Reviewer 5 Report
Dear authors,
the submitted paper present a deep study of the thermomechanical issues present in assembled microelectronic boards. The scientific approach is accurate and meticolous apart some minor errors, here listed:
A) line 17 and successive : "a dwell/ramp time of 15°/15 minutes"
From what I saw dwell is 15 mins and ramp is 15 °C/min. If you want to use the elegant combination better respect the ordering:
"ramp/dwell time of 15°/15 mins"
B) line 46-48, insert relative citations apart of this paper results. like fig. 3 comes from ? insert in the caption the relative citation
C) too many figures, fig. 4 is also in 14, here is useless compared to fig. 5, which textual dimensions are instead too small in character
D) Fig. 7 x-axis, please insert [min]
E) line 65-66, indicate TIM as terminal interface material
F) line 211 and last column table III
symbol ν (nu) is missing
G) fig. 13 it misses equivalent curve
of the equivalent material according to volume occupation
H) Comparative figure (main result) is definetely too little, it uses different scales (imperial or metric) and reversed colors, please correct
I) Fig. 15-16-17 shows CSED but it is not clearly indicated it refers to the maximum TCT temperature (120 °C) right ?
Lastly, I did not infer why the used thermal pad is a metal plate like in Fig. 6 and not a classic 3d one like in Fig. 2.
If possible better elucidate why and I suggest to make this research noteworthy in a future paper to combine thermal measurements of the chip self-heating w/wo the cooling element to look for the best trade-off between increased mechanical risks due to the added thermal plate and increased risks due to increased temperatures without adding thermal plate, also looking for thermal pad with intermediate sizes.
My best regards
Author Response

(The authors gave the same response as above.)
